# RegBR: A novel Brazilian government framework to classify and analyze industry-specific regulations

**Letícia Moreira Valle**[1]*, **Stefano Giacomazzi Dantas**[1], **Daniel Guerreiro e Silva**[1], **Ugo Silva Dias**[1], **Leonardo Monteiro Monasterio**[2,3]

**1** Department of Electrical Engineering, University of Brasília (UnB), Brasília, Brazil, **2** Institute for Applied Economic Research (IPEA), Rio de Janeiro, Brazil, **3** Brazilian Institute for Education, Development and Research (IDP), Brasília, Brazil

* leticia.valle@redes.unb.br

**Data Availability Statement:** As part of the RegBR's project visibility and transparency, all results and discussion on the created metrics are

## Abstract

Government transparency and openness are key factors to bring forth the modernization of the state. The combination of transparency and digital information has given rise to the concept of Open Government, that increases citizen understanding and monitoring of government actions, which in turn improves the quality of public services and of the government decision making process. With the goal of improving legislative transparency and the understanding of the Brazilian regulatory process and its characteristics, this paper introduces RegBR, the first national framework to centralize, classify and analyze regulations from the Brazilian government. A centralized database of Brazilian federal legislation built from automated ETL routines and processed with data mining and machine learning techniques was created. Our framework evaluates different NLP models in a text classification task on our novel Portuguese legal corpus and performs regulatory analysis based on metrics that concern linguistic complexity, restrictiveness, law interest, and industry-specific citation relevance. Our results were examined over time and validated by correlating them with known episodes of regulatory changes in Brazilian history, such as the implementation of new economic plans or the emergence of an energy crisis. Methods and metrics proposed by this framework can be used by policy makers to measure their own work and serve as inputs for future studies that could analyze government changes and their relationship with federal regulations.

## Introduction

In recent years, information and communication technologies (ICTs) have helped governments around the world to improve openness and transparency in their actions [1]. These aspects are key elements to increase trust in government, informed decision-making and democratic participation [2, 3]. The scope of e-government studies is expanding to consider not only government basic operations and service delivery, but also to enable citizen participation

available at https://infogov.enap.gov.br/regbr as well as the downloadable version of the centralized database built by the project.

**Funding:** This research was supported by the National School of Public Administration – ENAP, Government of Brazil (Grant TED UnB-ENAP 83/2018). There was no additional external funding received for this study.

**Competing interests:** The authors have declared that no competing interests exist.

and engagement using technology tools [4]. For instance, web applications, such as dashboards, are essential in bridging the gap between the government and the citizen [5, 6].

In the context of government regulations, ICTs provide new tools for governments to manage regulatory information, to advance public access to regulations, and to improve the transparency of the regulatory process. This is particularly important as the impacts and consequences of government regulations have been studied for decades, and they are considered a crucial policy tool for tackling market inefficiencies, [7] to foster economic growth and to develop a more prosperous society [8, 9].

In spite of recent advancements in Natural Language Processing (NLP) applications, their usage in the legal domain is still relatively under-explored. Some of the challenges in this area include the scarcity of relevant labeled documents, the cost of classifying these documents (often depending on a domain-expert such as a lawyer or law student) and the documents' length, typically longer than the standard length used for training NLP models, such as tweets, customer reviews, and other smaller documents. Despite these constraints, the application of machine learning techniques in the law domain is recently gaining ground.

For instance, [10] conducted a comparative study on the performance of various machine learning algorithms in classifying judgments of the Singapore Supreme Court written in English. Similarly, [11] presented results of machine learning algorithms in the task of predicting the field of law to which a case belongs.

Another common NLP application in the law domain is the prediction of court ruling decisions. For example, [12] used extremely randomized trees to predict the US Supreme Court's rulings and, more recently [13], tackled the task of predicting patent litigation and time to litigation. Finally, [14] proposed a model to predict the verdicts of the European Court of Human Rights (ECRH).

Regarding the regulatory field, an integrated approach that covers the management of regulations, efficient access, and retrieval of regulatory information is often lacking [15]. The creation of an information infrastructure that allows government agents and the general public to compare and contrast different regulatory documents will improve the understanding of regulations, and increase government transparency.

Some recent studies and projects are advancing this area of governance. One example of such work is RegData [16], where the authors quantified federal regulations by industry and by the regulatory agency for all federal regulations of the United States. This type of work is relevant as several relationships between industry regulation and economic interests can be drawn from analyzing data. Other English-speaking countries such as Canada [17] and Australia [18] also implemented a similar framework.

With this in mind, we propose a framework applicable to Brazil called RegBR, a multidisciplinary project, produced by Brazilian researchers in partnership with the National School of Public Administration (Escola Nacional de Administração Pública, ENAP), which aims to produce relevant information on the national regulatory situation. Instead of responding to a citizen's demand to access some information, RegBR already deliver information to the citizens in a simple and visually friendly way, centralizing information of different sourcers, compiling results and reducing access costs.

In addition to the direct use by the population, RegBR has several applications to the federal government. First, the framework and its data can subsidizes new regulatory studies. For instance, the Brazilian Public Service Journal (*Revista do Serviço Público*) opened a call for papers using RegBR as its data source.

Second, RegBR can assist regulatory agencies decision makers in measuring their own work, i.e. the tool allows the heads of regulatory agencies to measure what their organization produces in terms of volume and characteristics of regulations. It allows managers to have

concrete parameters to quantify and monitor regulations, such as: restrictiveness, measure of interest, influence of the regulated sector in the economy, and linguistic complexity of the regulations. In this context, the decision maker who wants, for example, to make some specific sector of the economy less regulated, can use RegBR to evaluate how the regulations produced by his organization behave over time.

Third, RegBR can be used as a comparative apparatus allowing the Brazilian Federal Government to compare its normative production by industry, by regulator and by metrics with distinct countries that already have similar metrics, like United States, Canada and Australia through RegData initiative for example. Lastly, the proposed framework achieves the proposal of the Brazilian Law 12.527 from 2011, the Access to information law, by ensuring access to data and fomenting active transparency to public information.

The active monitoring of Brazilian regulations is a high-priority activity as Brazil ranks at 46 from 48 countries evaluated by OECD in regulatory performance [19] and also in order to meet the criteria of the Brazilian Decree 10.139 from 2019, that establishes that every organization in the Federal Government must revoke normative that no longer had applicability with the goal of reducing the number of normative acts and make the legal system more efficient. New initiatives in regulatory performance will enable an institutional environment improvement, which could attract more foreign investment and improve the country's performance on international indicators such as the Product Market Regulation (PMR) indicators. Moreover, this framework will increase government transparency and the right to access public information, regarded as essential to democratic participation [1, 3, 20].

In this context, RegBR aggregates and processes data from different decentralized sources and applies Extract, Transform, and Load (ETL) techniques. We build data pipelines responsible for scraping and cleaning data from the official government websites of the leading regulatory agencies in Brazil to consolidate a novel database of federal legislation that we could use in our analysis.

Next, RegBR applied NLP techniques in order to classify federal legislation regarding their CNAE areas (in Portuguese *Classificacao Nacional de Atividades Economicas*, used to divide economic activities into different sectors.). As a consequence, RegBR also contributes to the literature of Portuguese NLP at large by benchmarking the different text classification techniques, such as Bag-of-Words (BoW) [21], Word Embedding [22], and Transfer Learning [23] against a set of the federal normative legislation corpus, therefore providing a blueprint to future developments in the NLP field in Portuguese.

Finally, yet another contribution from RegBR is the creation of metrics to evaluate regulations, such as their measure of interest, linguistic complexity and industry citation relevance. We also adapted the original restrictiveness metric from RegData [16] to incorporate additional restrictive words in Portuguese.

## Materials and methods

The RegBR framework is composed by two main layers. The first layer classifies regulatory texts in different sectors of the economy using machine learning models. This step is necessary as, for the most part, the Brazilian legislation is not labeled according to the economic sectors to which they regulate, and the legislative body is too large to be classified manually. The second layer uses this initial classification to apply different proposed metrics that measure different aspects of regulation, which can be monitored by different economic sectors, such as linguistic complexity, restrictiveness, citation influence and measure of interest over time.

In the context of normative acts text classification, we can formally define $d_i \in D$ as a document from a set of normative act texts $D = \{d_0, d_1, d_2, \ldots, d_n\}$ and $c_i \in C$ as a label from a set of

labels $C = \{c_0, c_1, c_2, \ldots, c_{18}\}$, which represents the eighteen different classes based on the Brazilian Institute for Geography and Statistics (*Instituto Brasileiro de Geografia e Estatística*, IBGE) economic sector classification. Hence, we define Text Document Classification (TDC) as the task of assigning $d_i$ to $c_i$ in order to structure dataset efficiently and accurately [24].

For the creation of metrics that allows regulators and policymakers to better identify and prioritize regulations that may need reform, we propose a set of metrics $M = \{m_0, m_1, m_2, m_3\}$, described as:

- Restrictiveness ($m_0$): indicates the regulatory restriction counts and how the sectors of the economy have become more or less regulated by more or less restrictive laws over the years. This metric is adapted from [16].

- Industry citation relevance ($m_1$): Calculates the relevance of regulations to economic sectors and industries, based on the frequency of citations of these sectors' keywords in the general context of normative acts. This metric is also adapted from [16], including modifications presented in Section 6.3 of this work.

- Measure of interest ($m_2$): Indicates how popular a law is for the population, based on the active search for that law on Google, and the frequency of citations of the law in the Official Gazette of the Federal Government. This metric is a novel contribution from RegBR.

- Linguistic complexity ($m_3$): Uses the median sentence length, the frequency of conditional terms, and Shannon's entropy to measure the linguistic complexity of a document. These metrics are adapted from [17].

This section presents the implementation details of the ETL pipeline used to aggregate and process regulatory texts from different distributed sources. The consolidated data is made available for researchers and the general public, which is an important step towards government accountability and transparency.

## Database modeling and ETL routines

The RegBR database was designed to store information regarding the economy sectors classification and the creation of general metrics. The collection of all federal regulatory acts in a single, centralized, and automated database is one of the main RegBR contributions as it reduces the barrier to data release [25] and increase government transparency. The compiled database will be available for other researchers who want to use the Federal Regulation data to do research or other related activities.

One of the main challenges in implementing this ETL pipeline is that Brazilian regulatory norms are not centralized in a single source. Before the collection and extraction of federal laws in a decentralized manner, a study was carried out to verify if any centralized databases already exist for this matter. A project called LEXML [26], of the Brazilian Federal Senate, was considered, but since it only provides metadata, it could not be incorporated on RegBR, once it needs the normative acts' full text. In addition, the output results do not aid in the development of this work, as the results from LEXML are not structured the results by normative act type.

Due to the current limitations of data access and sources decentralization, we implemented an ETL pipeline to deal with different sources organization using scraping scripts and managing the data collection routines. For this purpose, we employed tools such as Python [27], with the use of the *Beautiful Soup* [28] and Selenium [29] libraries, and the Apache Airflow [30] platform, which is an open-source workflow management platform that can run tasks based on a defined schedule (for example, hourly or daily) or based on external event triggers, which

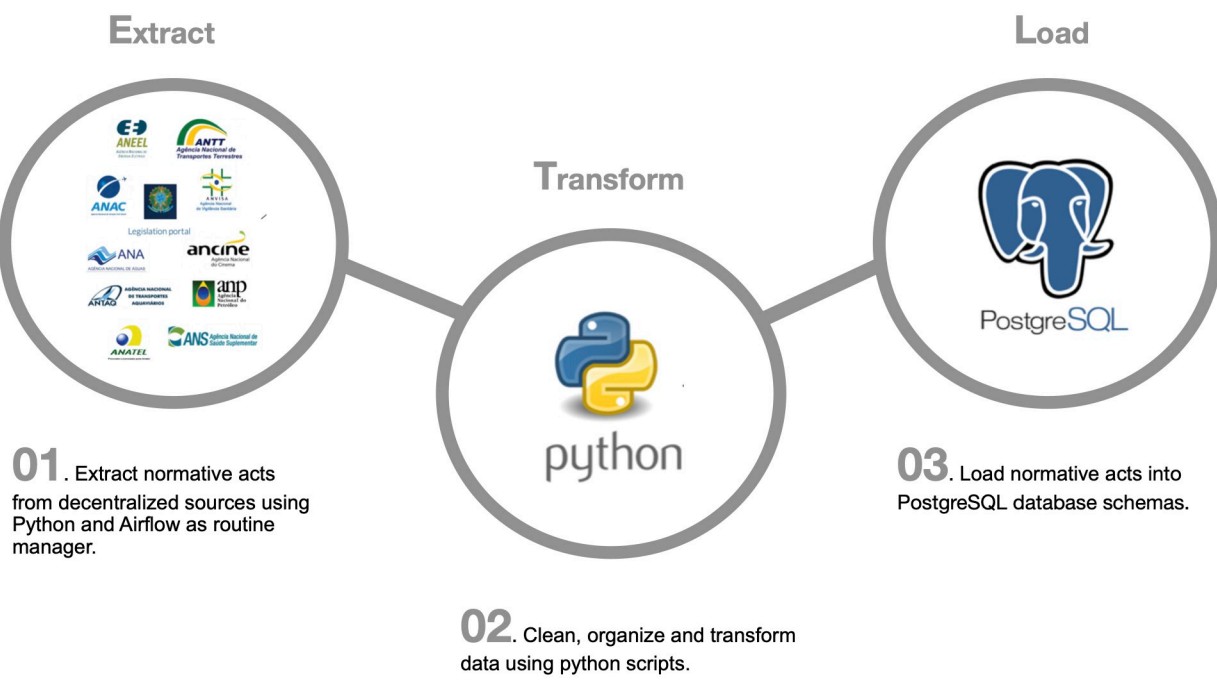

**Fig 1. RegBR ETL pipeline.**

allows the automation of the database updates. Fig 1 summarizes the general structure built to carry out the ETL procedures.

## Dataset annotation

Gathering a labeled dataset is an essential procedure for classifying normative acts into the different economic sectors that they regulate using text classification techniques.

The National Classification of Economic Activities (*Classificação Nacional de Atividades Econômicas*, CNAE) [31] is the official categorization, adopted by IBGE, for the production of statistics for different economic activities and by Public Administration. CNAE is similar, but not identical, to the North American Industry Classification System (NAICS) or the *Nomenclature Statistique des Activités Économiques dans la Communauté Européenne* (NACE).

The CNAE is structured in twenty-one main categories, called sections, which in turn have four additional hierarchical levels: division, group, class, and subclass. The fifth level, designated subclass, is defined for use by the Public Administration. Fig 2 shows an example of the CNAE structure for the 'Human health and social services' sector.

| Hierarchy | | |
|---|---|---|
| Section: | Q | Human health and social services |
| └ Division: | 86 | Human health care activities |
| └ Group: | 86.1 | Hospital care activities |
| └ Class | 86.10-1 | Hospital care activities |
| └ Subclass: | 86.10-1/01 | Hospital care activities, except emergency room and emergency care units |

**Fig 2. CNAE's structure example—Human health and social services.**

**Table 1. Final classes.**

| Class | Definition |
|-------|------------|
| 1 | Agriculture, livestock and forest |
| 2 | Extractive industry |
| 3 | Transformation industry |
| 4 | Electricity and gas |
| 5 | Water, sewage and waste |
| 6 | Construction |
| 7 | Commerce, accommodation & food and real estate services |
| 8 | Transportation, storage and mail |
| 9 | Information and communication |
| 10 | Financial, insurance and related services |
| 11 | Professional, scientific and technical activities |
| 12 | Administrative activities and complementary services |
| 13 | Public administration, defense and social security |
| 14 | Education |
| 15 | Human health and social service |
| 16 | Arts, culture, sports and recreation |
| 17 | Other services |
| 18 | Non-regulatory |

For project scope reasons, only the first hierarchy level is used in this work, resulting in 21 different classes of economy sectors. Moreover, for the purposes of this project, some classes were not relevant as they present low frequency and/or are very similar to other sectors. Thus, in order to simplify the classification process, we decided to perform the following aggregations:

- Classes 7 (trade and repair of vehicles), 9 (accommodation and meals) and 12 (real estate activities) of the CNAE were merged into 7: Commerce, Accommodation and Food, and Real Estate Services.

- Classes 19 (Other service activities), 20 (Domestic services) and 21 (International organizations) were merged into 17: Other services.

In addition to these aggregations, we added an extra class (18) to indicate legislation that is not related to regulatory activities. The S1 Table presents the complete table with the 18 final sections together with their divisions.

In this paper, we use class and economy sector interchangeably. The dataset annotation was carried out by a consultant with expert domain knowledge following the classes definition illustrated in Table 1, that presents a simpler view of the 18 final classes considered for classification. The labels in the annotation process were defined according to the main economic sector affected by each normative act analyzed.

The norms types defined as *Ordinances* and *Resolutions*, which correspond to about 60% of the total normative set, did not need to be annotated and classified since the regulatory agency that regulates the normative act is known, as well as its economic area of activity. From the remaining normative acts which needed to be classified, 20% of it was annotated by the consultant in order to create the training and test dataset.

## Details on text classification

Over the last few decades, especially with recent breakthroughs in NLP and text mining, text classification applications have been widely studied and implemented [32]. One area that has

grown and gained relevance in recent years is related to the classification of legal texts [10], which usually have complex and technical language, imposing manual classification tasks for a select group of jurists with specific domain knowledge. Moreover, legal texts are composed of large amounts of words and content, making it infeasible to perform the classification task manually and efficiently.

The way a problem is modeled has a strong influence on the performance of the learning system trying to solve it. For text categorization problems, the method used to transform words into a numeric representation suitable for the classifier is crucial to determine its efficiency. This article applies three approaches to represent text data: Bag of Words (BoW), Word embedding, and Transfer Learning, each presenting different characteristics. We evaluated these three approaches by increasing complexity and recentness.

**Statistical models.** We employed different classifiers in order to evaluate the performance of base models. The words were transformed in features using TF-IDF, and the classifiers were implemented using Scikit-learn [33]. We selected some of the most commonly used classifiers in machine learning applications such as the Logistic Regression, Support Vector Machine (SVM) with linear kernel and Gradient Boosting Classifier [34]. In addition to these classifiers, we also evaluated the Ridge classifier, which is a models that converts the task into a regression and generates labels accordingly. Due to their expressive power, we also included a fully-connected (FC) neural network using the TF-IDF features. Finally, we also include an SVM classifier that uses features extracted using Latent Semantic Analysis (LSA) [35]. According to [10], this combination is still a classical approach in the legal text classification literature.

Since our dataset was fairly imbalanced, i.e., the different classes are not approximately equally represented, we also use the SMOTE [36] oversampling technique in order to improve the performance by generating samples from minority classes.

The hyperparameters were chosen after using grid search for each classifier. This process was also used to select the best parameters for the TF-IDF vectorizer. More specifically, the maximum number of features, the cut-off frequencies, and the *n-gram* range.

An alternative vectorizer (*Count-Vectorizer*) was also tested but performed consistently worse than TF-IDF. This also happened when we tried balanced class weights (giving more weights to less frequent classes when evaluating the loss function), but the results were also underwhelming.

Moreover, the SMOTE oversampling only improved the performance of the logistic regression classifier. For the LSA + SVM classifier, we also tried a different number of topics, but it turned out that 500 was the optimum number of topics.

**Word embedding models.** The use of word vectors pre-trained on large corpora have proven to capture syntactic and semantic word properties. We leverage this capability in our application by using *word2vec* [22] and *GloVe* [37]embeddings, both with 300 dimensions, pre-trained on a Portuguese corpus [38].

These embeddings were used by two different neural network classifiers; a Convolutional Neural Network (CNN) [39] and a Long Short Term Memory (LSTM) network [40]. These architecture were chosen as they are often used in NLP applications [41] and are also some of the building blocks of more complex deep learning models.

**Transfer learning models.** Recently, impressive results were achieved by language models that were pre-trained on large unlabelled corpora and then fine-tuned for specific tasks. This transfer learning method can be advantageous in tasks with a lack of labeled data, such as in the legal domain. Therefore, we evaluated two variants of BERT [23] known as $BERT_{base}$ (12-layers; 110M parameters) and $BERT_{large}$ (24-layers; 340M parameters).

**Table 2. Hyperparameters used to generate the results.**

| Models | Hyperparameters |
|---|---|
| Tf-idf vectorizer | Maximum number of features = 10,000 / n-gram range = (1,1) /no constraints in the frequency of words |
| Logistic Regression (LR) | Regularization coefficient = 2 |
| Ridge Classifier (RC) | Regularization coefficient = 1 |
| Oversampling | Over-sampling using SMOTE and cleaning using Tomek links with default parameters |
| SVM | Linear Kernel, Regularization coefficient = 0.5 |
| XGBoost | XGBoost with 100 estimators |
| SVM + LSA (500) | LSA with 500 topics / SVM with linear kernel with regularization coefficient = 1 |
| Word2vec | 300 dimensions |
| GloVe | 300 dimensions |
| LSTM | Embedding layer followed by a bi-directional with 64 hidden unitsfollowed by a fully connected layer with 256 units with dropout = 0.1and a final layer with 64 hidden units. |
| CNN | Embedding layer followed by 4 convolutional layers with 36 filters ofvarying kernel size (1,2,3,5) followed by a fully connected layerwith 144 hidden units with dropout = 0.1 |
| BERT | BERT pre-trained (12 layers with 110 million parameters for the base model,and 24 layers with 335 million parameters for the large model)using Portuguese corpus [44], followed by a final fully connected layer with dropout = 0.05 |
| ULM-FiT | One AWD-LSTM layer [45] followed by 4 QRNN layers [46]with dropout and a final fully connected layer |
| NN + TF-IDF | Fully connected network with 25 hidden units in the first layer followed bybatch normalization, dropout = 0.25 and a final layer with 18 units |

However, one of the limitations of BERT is the self-attention transformer architecture [42] which only accepts up to 512 tokens. Since some legal texts are longer than this, we also employ the ULM-FiT model [43], which accepts longer inputs due to its stacked-LSTM architecture.

In order to fine-tune BERT to our problem, we used the PyTorch pre-trained implementation in Portuguese [44] and added a final layer with the correct dimensions.

For each studied approach, their respective hyperparameters are presented in Table 2.

It is well known that tuning hyperparameters can be time-consuming, as different methodologies could be applied to optimize their selection [47]. Therefore, we performed a grid-search using values that are found in the literature for each one of the models. The final values were picked according to the highest accuracy and f1-score in a 5-fold cross validation.

## Normative acts data structure

The corpus comprises 112,000 normative acts of the Brazilian federal legislation written in Portuguese, divided in eight types, since 1891. In this context, a normative act means any law, decree, resolution, regulation, administrative direction, instruction, rule, ordinance, or other decision that creates legal consequences for more than one individual. Table 3 presents all types of normative acts studied in this work and the corresponding translation to English.

The first six types come from the Brazilian Legislation Portal [48] in HTML format, while the remaining two normative acts types come from decentralized sources of the electronic portals of the 11 Brazilian regulatory agencies in various formats, including PDF, HTML, and images. The median length of a normative act is about 469 tokens, which is significantly longer than the typical customer review or news article commonly found in datasets for benchmarking machine learning models on text classification.

The dataset contains about eight thousand labeled acts divided into 18 classes. It includes all legislation available on the internet from the cited sources with normative acts since the end

**Table 3. Normative acts information.**

| Type | Normative act in Portuguese | Normative act in English |
|------|------------------------------|---------------------------|
| 1 | *Emenda Constitucional* | Constitutional Amendment |
| 2 | *Lei Ordin*á*ria* | Laws |
| 3 | *Decreto-lei* | Decree Law |
| 4 | *Medidas provis*ó*ria* | Provisional measure |
| 5 | *Lei complementar* | Supplementary Law |
| 6 | *Decreto* | Decree |
| 7 | *Resolução* | Resolution |
| 8 | *Portaria* | Ordinance |

of the 19th century. For information, we split the labeled data into 75% training and 25% testing.

## Results and discussion

### Normative acts text classification

We present the text classification task results in two parts: first using the entire set of normative acts starting with the first normative act available on the digital platforms used as source, in 1891; and soon after, using only normative acts from 1964 onward in order to delimit a clear and more linguistically homogeneous temporal scope, since the vocabulary used at the beginning of 20th century is considerably different from the current one when we refer to legal texts.

Another reason for picking 1964 is that it represents a milestone in Brazilian political and economic history. A military dictatorship was established in that year, and many normative acts remain valid, even after the 1988 Constitution. This year was also chosen among several tested years for presenting the best trade-off between including the most normative acts in the training phase and also having a consistent linguistic style between the text analyzed. Tables 4 and 5 evaluate models using all data, and data after 1964, respectively.

**Table 4. Classification results with all data.**

| Models | Accuracy | Average F1-score |
|--------|----------|------------------|
| Logistic Regression (LR) | 62.64 ± 1.03% | 0.571 ± 0.093 |
| Ridge Classifier (RC) | 63.77 ± 0.94% | 0.59 ± 0.006 |
| $LR_{SMOTE}$ | 63.57 ± 0.83% | 0.597 ± 0.009 |
| SVM | 63.96 ± 1.19% | 0.592 ± 0.013 |
| XGBoost | 60.94 ± 0.79% | 0.553 ± 0.013 |
| $Ensemble_{RC,LR_{SMOTE}}$ | 63.59 ± 0.82% | **0.598 ± 0.09** |
| $Ensemble_{RC,SVM}$ | **64.06 ± 1.18%** | 0.592 ± 0.10 |
| $SVM_{LSA}$ | 59.50 ± 4.7% | 0.538 ± 0.045 |
| $LSTM_{word2vec}$ | 57.08 ± 1.02% | 0.5043 ± 0.03 |
| $CNN_{word2vec}$ | 59.99 ± 0.47% | 0.541 ± 0.072 |
| $LSTM_{GloVe}$ | 57.48 ± 0.38% | 0.5151 ± 0.088 |
| $CNN_{GloVe}$ | 59.48 ± 0.52% | 0.543 ± 0.064 |
| $BERT_{base}$ | 61.84 ± 0.85% | 0.551 ± 0.024 |
| $BERT_{large}$ | 48.70 ± 1.19% | 0.382 ± 0.067 |
| ULM-FiT | 55.29 ± 1.03% | 0.526 ± 0.055 |
| FC Neural Network$_{TF-IDF}$ | 58.58 ± 0.9% | 0.541 ± 0.011 |

**Table 5. Classification results with data post-1964.**

| Models | Accuracy | Average F1-score |
| --- | --- | --- |
| Logistic Regression (LR) | 65.93 ± 1.25% | 0.575 ± 0.015 |
| Ridge Classifier (RC) | 67.97 ± 0.95% | 0.612 ± 0.015 |
| LR$_{SMOTE}$ | 66.15 ± 0.011% | 0.609 ± 0.013 |
| SVM | 67.72 ± 1.31% | **0.619 ± 0.013** |
| XGBoost | 63.91 ± 1.11% | 0.568 ± 0.015 |
| Ensemble$_{RC,LR_{SMOTE}}$ | 64.96 ± 1.83% | 0.591 ± 0.022 |
| Ensemble$_{RC,SVM}$ | **67.97 ± 1.09%** | 0.616 ± 0.015 |
| SVM$_{LSA}$ | 61.51 ± 3.94% | 0.531 ± 0.031 |
| LSTM$_{word2vec}$ | 58.37 ± 1.02% | 0.521 ± 0.012 |
| CNN$_{word2vec}$ | 61.66 ± 0.92% | 0.565 ± 0.079 |
| LSTM$_{GloVe}$ | 59.78 ± 1.36% | 0.533 ± 0.014 |
| CNN$_{GloVe}$ | 61.21 ± 1.57% | 0.565 ± 0.016 |
| BERT$_{base}$ | 62.21 ± 0.94% | 0.514 ± 0.061 |
| BERT$_{large}$ | 52.72 ± 0.89% | 0.428 ± 0.056 |
| ULM-FiT | 58.14 ± 0.92% | 0.538 ± 0.033 |
| FC Neural Network$_{TF-IDF}$ | 63.66 ± 0.94% | 0.569 ± 0.020 |

Across the models implemented, word embedding models consistently under-performed the statistical and transfer learning models on accuracy. Regarding the F1-score, word embedding models performed worse, on average, than the statistical models. This underwhelming results could be explained by the corpus used for pre-trainining the embedding. The Brazilian Portuguese and European Portuguese documents used come from different sources and from USP Word Embeddings Repository [38], not from a specific and targeted set of legal and bureaucratic texts from a federal government.

When we analyze transfer learning approaches, BERT models performed slightly worse than the best statistical models, and ULM-FiT had the worst performance overall, both accuracy and F1-score. Again, the causes of inferior performance with respect to statistical models could be attributed to the generality and the small number of pre-trained language models in Portuguese. We used Neuralmind [44] BERT language model and FastAI ULM-FiT based language model, both trained in a wide range of texts but not necessarily linked to legal texts.

Most surprisingly, statistical models emerged as the best performing approaches on both accuracy and macro averaged F1 scores. Statistical models are significantly faster for training and testing when compared to the implementations using deep neural networks. Due to the lower computation cost, it is also possible to combine different models as an ensemble, which often perform better than any of its single classifiers [49].

Although considering only post 1964 material to train and test the model makes us lose about 40% of the normative acts, an improvement in the performance metrics of them is noticeable, with both accuracy and F1 score metrics improving. This improvement may be partly explained by the fact that the language observed in the texts is more homogeneous.

Additional analyzes were performed to handle the fact that the dataset was unbalanced. Two different methodologies were used in order to obtain similar numbers of examples for the different classes. First, an undersampling method was applied by randomly eliminating examples from the most numerous classes. The performance obtained was not satisfactory, possibly due to the decrease in the training dataset. Then, the dataset was balanced using SMOTE [36], which oversampled the examples of the least represented classes. In this case, the results were

slightly worse than the classification using the unbalanced data, indicating limitations of the technique for the problem at hand.

In fact, we expected transfer learning models, being state-of-the-art on many non-legal NLP tasks, to perform best here. The results obtained with the statistical models, around 68% of accuracy and 62% of F1-score, for a larger number of classes, is an exciting result and is similar to what was shown in other legal text classification benchmarks in other languages [10]. One of the limitations of the BERT model is the fact that this model is not capable of working with texts larger than 512 tokens, which, in our case, represents more than half of our database. This fact can help explain the slightly worse performance compared to some of the other simpler models we tried. It is possible that different architectures of the BERT family, which may have the capacity of working with longer texts, could exhibit better performance. This investigation might be explored in future work.

The fact that simpler classifier outperformed complex deep neural networks helps to illustrate the importance of benchmarking different models and techniques instead of just using the latest model available. This analysis is even more important in NLP applications, where the language and context used to pretrain the model can have significant impacts on their performance.

Due to its good results and low computational cost, we employ the $\text{Ensemble}_{RC,SVM}$ as the final model to classify regulations into different economic sectors. Moreover, this model also allows us to verify the most relevant terms in classifying each sector, which improves the model's transparency and interpretability.

## Regulatory metrics conception

This section presents metrics that allow regulators and policymakers to better identify and prioritize regulations that may need reforms. In this sense, RegBR provides a variety of quantitative data and indicators, including

- Regulatory stock analysis over time, making the federal regulatory flow transparent;

- Restrictiveness metric, indicating the regulatory restriction counts;

- Industry citation relevance metric, which calculates the frequency of industry-relevant terms in the context of federal regulations among the economic sectors considered;

- Measure of interest, indicating how popular a law is for the general population and for the federal government;

- Linguistic complexity metric, measuring the linguistic complexity of a normative act.

All the metrics and indicators presented in this section are calculated based on legislative documents obtained via the automated ETL routines presented in the section Database modeling and ETL routines and classified into sectors of economy using the best performing model presented in section Results and Discussion. The ETL system and text classification task are updated every 3 months to keep the project results up-to-date and available to the public and decision makers to use.

**Regulatory stock analysis over time.** The concept of *Regulatory stock Management* is not new in the Brazilian government context. Nonetheless, it gained more relevance since the publication of Decree 10.139, of 28 November 2019, which imposes the review and consolidation of all normative acts with a hierarchy lower than the decree by the end of 2021. In addition, as determined by the decree, each federal administration body and entity must divide all its normative acts by thematic relevance and review them by steps [50].

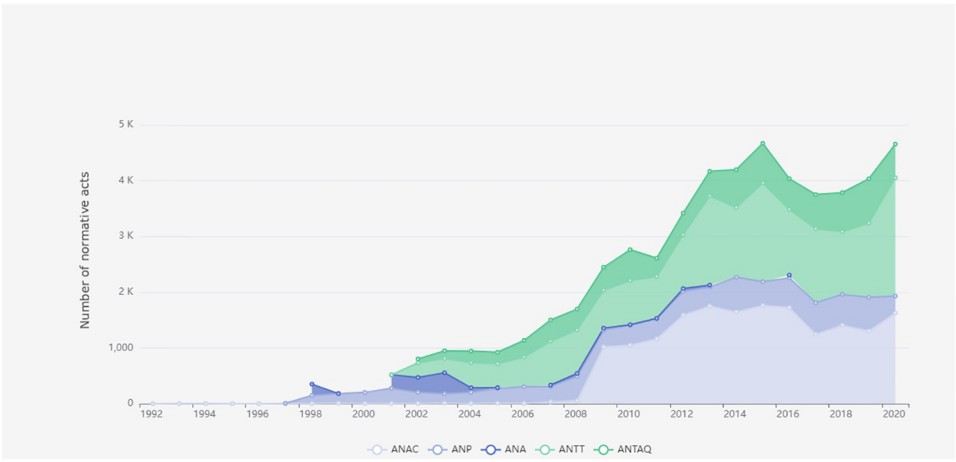

**Fig 3. Normative acts of regulatory agencies over time.**

In order to adapt to this new context, some regulatory agencies are establishing work groups for quantitative mapping of regulatory stock. RegBR brings a general analysis of the Brazilian regulatory stock filtered by sector of the economy or by regulatory agency, to assist regulatory authorities in managing the country's regulatory stock and to better adequate the regulatory process to international quality parameters.

Fig 3 illustrates the quantitative analysis of the normative acts *Resolutions* and *Ordinances* by the main regulatory agencies in terms of volume. In this Fig, we can see and monitor what their organization produces in terms of volume and what are their trends. It is possible to observe that the volume of new regulations is closely related to the the creation of relevant Brazilian regulatory agencies that took place in the early 2000s.

In the same context, Fig 4 presents the volume of normative acts filtered by the main types of normative acts. The surge of new ordinary laws at the end of the 1980s can be linked to the Brazilian Constituent Assembly that established the current Brazilian Constitution in 1988.

The information presented in the aforementioned figures can be filtered by economic sector and normative act situation and is available for public consultation, which increases

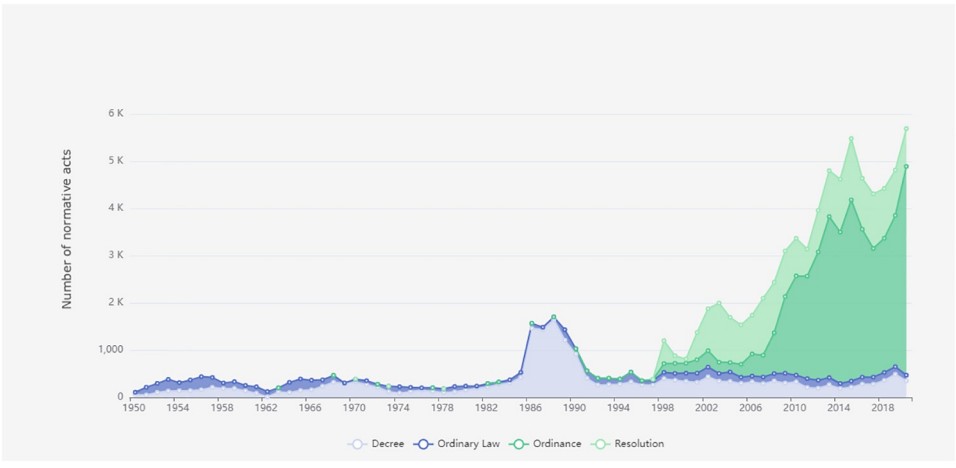

**Fig 4. Regulatory stock evolution over time.**

government transparency and allows easy access to information for citizens and policy makers interested in monitoring their work metrics.

**Regulation restrictiveness metric.** Recently, regulatory reforms have gained increasing attention in the political and economic context [51], and, as a consequence, researchers have tried to introduce simple, direct, and straightforward forms to quantify regulations and perform ex-post evaluation [52].

One of the first methods used for this purpose was based on page counts [16]. Due to its simplicity, this method does not always correctly represent the complexity or importance of the laws since long texts are not necessarily stricter than short and concise texts. In addition, the fact that some texts use more tables, graphs, diagrams, and annexes, which disproportionately increases the number of pages, can negatively influence the use of this metric as the primary method for quantifying the regulatory power of a law.

In order to overcome the problem of different text formatting [53], proposed the use of file size data for quantification purposes. However, the presence of large graphics and tables can still bias this measure.

As a methodology capable of overcoming these problems, RegData US [16] proposed the use of word count to quantify the restrictiveness of a piece of regulation.

Regulatory restrictions are defined as words and phrases in a regulatory text context that indicate specific obligations or prohibitions [54]. As normative texts are intended to restrict or expand legal scopes, these texts often use certain verbs and adjectives such as 'shall' and 'must'. The restriction metric is then measured by the total number of occurrences of restrictive words in a set of laws within the body of the normative act.

In the Brazilian context, we have seen a greater number of enacted normative acts in the last few years. For this reason, RegBR proposes a slight modification in the original metric of law restrictiveness of RegData US. In addition to counting the restrictive words in a set of normative acts, we divide this number by the number of normative acts in the set, by economic sector in each year, thus obtaining a metric for the average number of restrictive words by normative acts over the years for each economic sector. Hence, we can understand whether the average number of restrictive words by normative acts has increased, decreased, or remained constant over time. Thus, the regulation restrictiveness metric for RegBR is defined as

$$\text{restrictiveness}(\text{year}, \text{ economic sector}) = \frac{\sum \text{ restrictive word count}}{\sum \text{ law}}, \tag{1}$$

in which the word counts are defined by a list of restrictive words in Portuguese that intend to restrict or expand legal scopes. This list contains the words *vetoed, forbidden, closed, prohibited, denied, determines, obliges, orders, imposes, limits, delimits, demarcates, restricts, confines, reduces, defines, must, shall, needs*. It was proposed by the authors and validated by law professionals with vast experience in the legislative field.

After generating the time-series with the average number of restrictive words per law and per year for each economic sector, we perform a stationarity statistical test of the restrictiveness metric over time to assess the existence of trends in the data [55, 56]. This would indicate possible increases or decreases in the average strictness of the laws over time. For this purpose, we applied the Augmented Dickey Fuller (ADF) test [57] as well as the Kwiatkowski-Phillips-Schmidt-Shin (KPSS) test [58] together [59]. In general, if the results of both tests suggest that the series is stationary, we can consider its stationarity with high confidence [59]. In simple words, we may infer whether the mean function of the series is constant or not.

As a result of the experiment, 10 out of 18 economic sectors tested to be non-stationary, presenting a positive trend of increasing the number of restrictive words over the years:

Agriculture (1), Transformation industry (3), Electricity and gas (4), Water and sewage (5), Information and communication (9), Finance (10), Professional Scientific activities (11), Administrative activities (12), Education (14) and Human health (15). Fig 5 shows the restrictions word count per law over time, since 1964, for five of the economic sectors discussed in this work in a cumulatively and non cumulatively way.

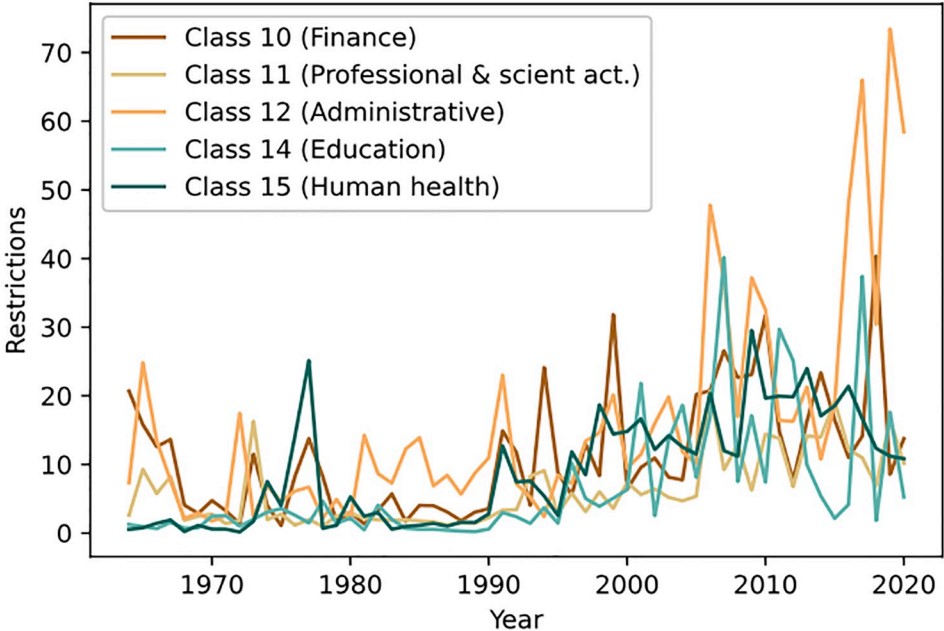

(a) Brazilian restrictive word count per law, 1964 - 2020.

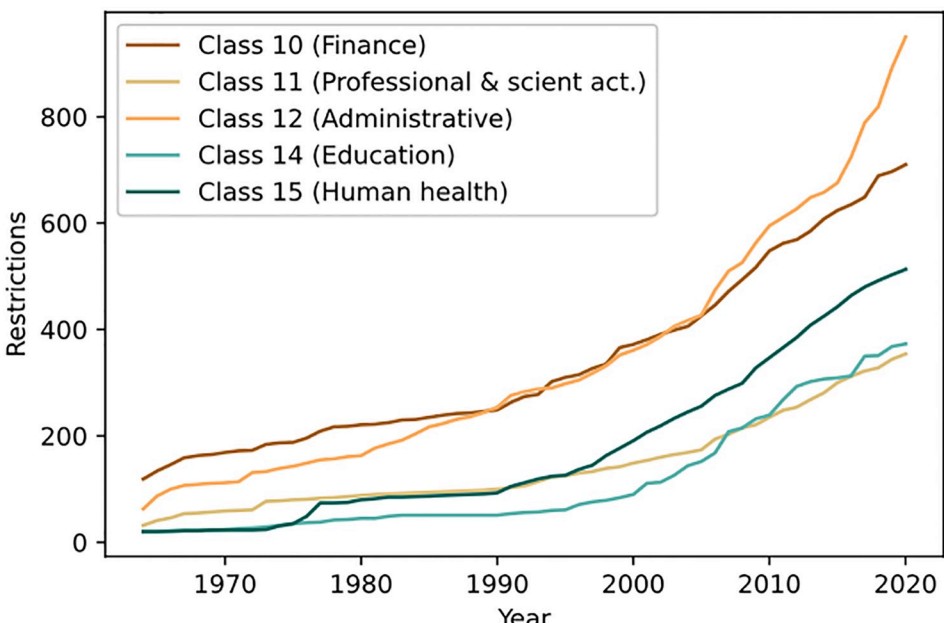

(b) Brazilian restrictive word count, cumulatively, 1964 - 2020.

**Fig 5. Brazilian restrictive word count.**

**Industry citation relevance.** The Industry citation relevance metric measures the influence of the CNAE's economic sectors, also called industries, based on their citation frequency in the general corpus of normative acts. If words directly related to a particular economic sector are used frequently throughout the entire corpus, that sector is understood to show more relevance than an economic sector that is not frequently cited, which may indicate the sectors that have been prioritized in the context of regulatory legislation.

In order to calculate the industry citation relevance metric by year, for each industry we sum the total occurrences of the specific strings terms that represent the industry in that year and divide by the total number of words in the normative acts corpus from that specific year as

$$\text{relevance(year, industry)} = \frac{\sum \text{ industry specific strings on corpus}}{\sum \text{ corpus words}}. \tag{2}$$

Then, we normalize the metric values by dividing them by the maximum value in that year to obtain a range between 0 and 1, where 1 represents the most relevant sector in that year.

The industry specific strings terms were derived from the most relevant words for the RC + SVM text classification model, the top-accuracy performing method, as seen in Section Results and Discussion. The top 10 words that represent each industry are unique, i.e., they do not affect the metric for other industries because there is no overlap between different sectors.

Next, Fig 6 displays the final occurrences of industry-specific string terms and the final count of normative acts compiled for each industry studied for the year of 2020. For example, for the Transportation industry, we count about 35 thousand normative acts by this year and approximately 300 thousands occurrences of transport related terms.

In sequence, Fig 7 presents the industry citation relevance metric for the year 2020, calculated as the ratio between the number of occurrences of the industry-specific search strings and the total word count for 2020. In that year, the most relevant industries in the context of regulatory acts are 'Transport', 'Electricity' and, curiously, the 'Health' sector was only in 5th place, even during the COVID-19 pandemic.

In addition, we can check the relevance metric values for each industry over the years. To illustrate this, Fig 8 shows the relevance metric for three economic sectors since 1964. Only

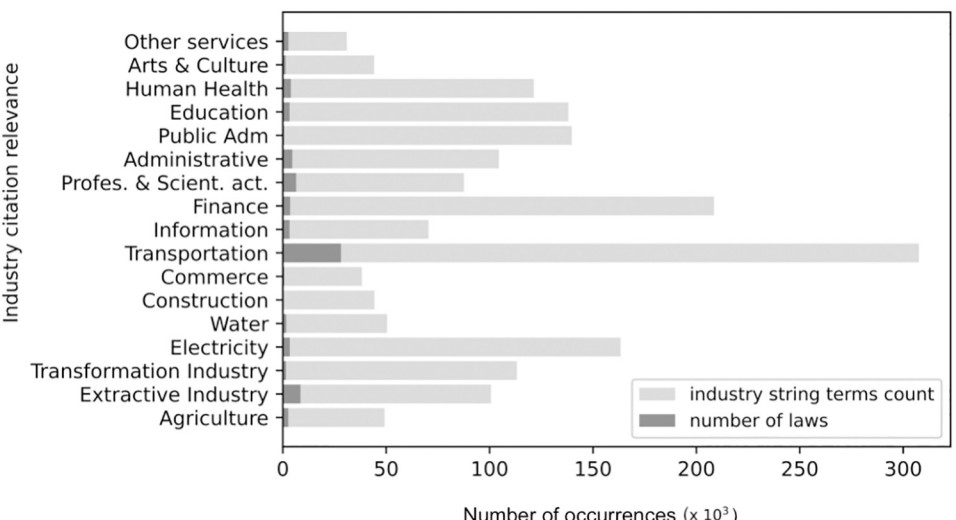

**Fig 6. String terms count.**

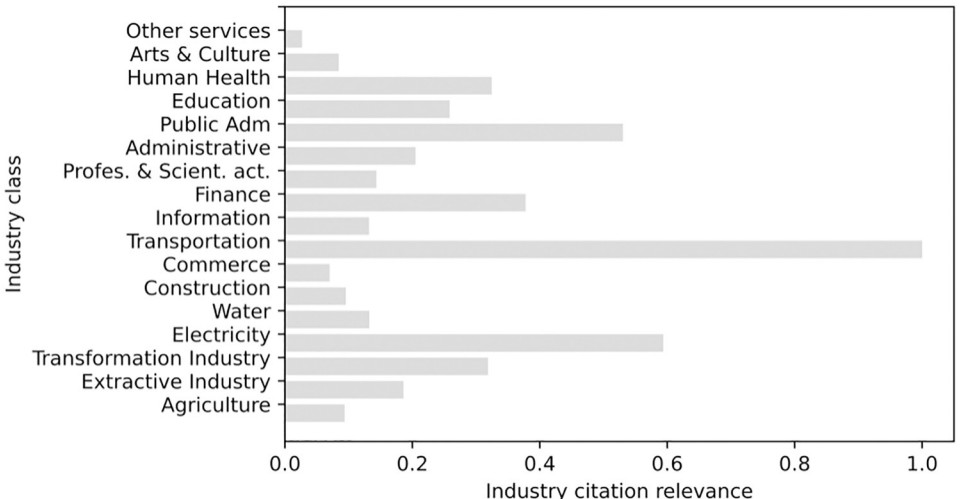

**Fig 7. Industry citation relevance metric for the year of 2020.**

three sectors are shown in the example in order to facilitate the visualization of the curves, but all data and metrics are available on the RegBR portal.

The industry citation relevance presented in Fig 8 correlates with some historical events described as follows. For instance, it is interesting to observe that between 2001 and 2004, due to the frequent energy blackouts and need for energy rationing, a regulatory reform was initiated in the Brazilian electrical sector [60], including the creation of the National Electric Energy Agency (ANEEL), which led to the increase in the relevance of this sector in the federal normative context in the aforementioned period, as shown in Fig 8 on class 4, Eletricity.

On a similar note, the increase shown in Fig 8 in transport industry citation relevance since 2000 correlates with the creation of the National Land Transport Agency (ANTT) and the National Waterway Transport Agency (ANTAQ) both in 2001, and the creation of the National Civil Aviation Agency (ANAC) in 2005, strongly regulating the transport sector in Brazil.

In the opposite direction, in the 80s and 90s, the Brazilian economy was marked by crises and hyperinflation. In 1986, the president at the time José Sarney launched the Cruzado Plan, the country's largest economic stabilization plan at that time. Several economic measures such as currency changes and freezing of wages, prices, and exchange rates were taken during the same year. With the return of hyper-inflation months later, several other plans were implemented until economic stabilization in the late 1990s as a result of the Real Plan [61], the thirteenth economic plan for stabilizing the Brazilian economy since the early 1980s, implemented by the Itamar Franco administration in 1994. Fig 8 shows that during these two decades, the finance sector was very relevant in the federal normative context, decreasing its relative relevance from the 2000s.

This type of analysis can help citizens to assess government's priorities, increasing transparency. For example, after a pandemic, one would expect an increase of industry citation relevance on the human health sector. By looking at how the metric of one sector compares to others, the population can make the government accountable for its prioritization.

**Normative act measure of interest.**   In order to allow the government to gain insights on regulatory topics of interest of its population and its internal administration, we also propose a novel metric, which is the measure of interest. This new metric indicates how popular is a

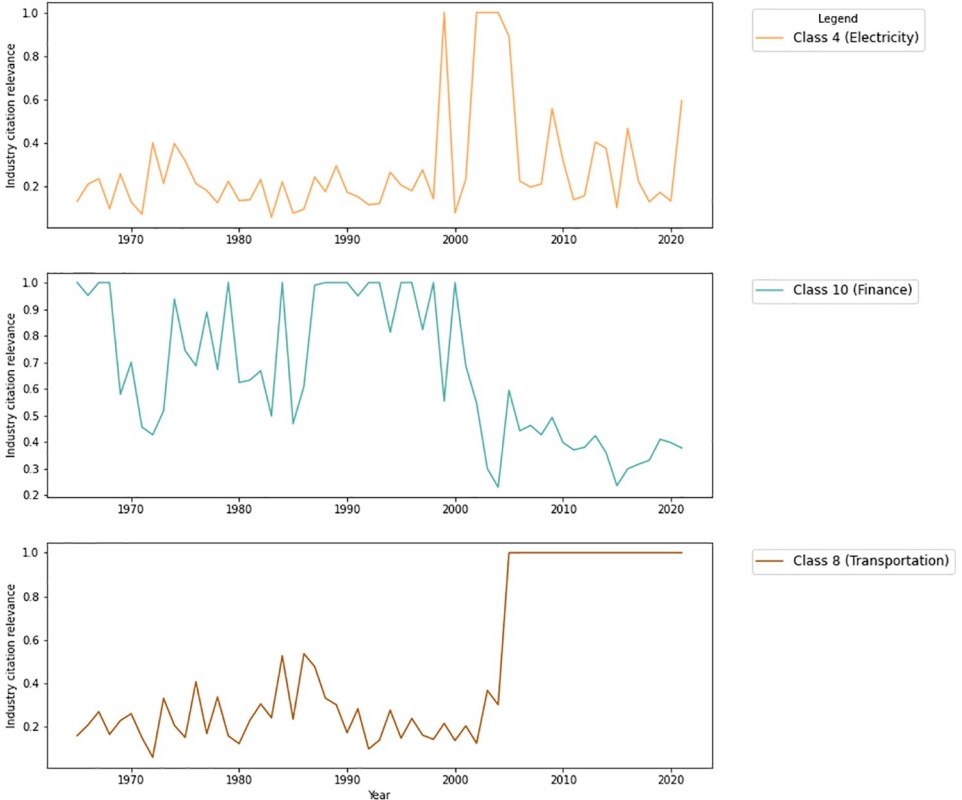

**Fig 8. Industry citation relevance for eletricity, finance and transportation economic sectors.**

normative act concerning a specific group. This metric aims to indicate which are the most popular normative acts, and consequently, the ones that generate the most interest from the general population and from the federal government.

The measure of interest is calculated based on the population active search for specific normative acts on Google, essentially the only search engine used by Brazilian people. In this context, we gathered information from the Google Trends engine alongside rules to search for normative acts and their search frequencies.

Google first launched Google Trends in 2006 to analyze the popularity of top search topics in the Google Search platform, across various regions and languages starting with data from 2004 [62]. This tool has been used in different applications over the years, such as prediction of the stock market behavior [63] and tourism patterns [64]. It was also used to understand the behavior of epidemiological diseases [65] and to calculate search interest of professional cycling [66].

Google Trends measures search interest in relative terms based on a randomly drawn sample, normalizing search data to make comparisons between different terms easier. In order to calculate this measure of interest, each data point is divided by the total searches of the geography and time range it represents to compare relative interest. Google Trends search interest ranges from 0 to 100, and it only shows data for popular terms, so search terms with low volume are set to 0.

For RegBR, the search strings used to calculate the Google Trends measure of interest were formed by the following elements: the act type followed by its number, a slash, and publication year. This is the most usual way of researching a normative act on search tools. Examples of

the search strings are *Law 8.112/1990* and *Constitutional Amendment 20/1998*. Normative act measure of interest was calculated for the first six normative acts types, excluding 'Ordinances' and 'Resolutions' as this search strings are not standardized with the other normative acts types.

As search parameters, we used 'BR' as geo-attribute and the last ten years as timeframe. It is worth mentioning that for normative acts existing for less than ten years, only the actual existence of the normative act was taken into account in calculating their average interest.

The Fig 9 shows the average interest for the 15 more popular normative acts. Not surprisingly, the most popular normative acts for the general public are connected with current social-economic aspects such as the COVID-19 pandemic, labor regulations reform, the statutes for people with disabilities, micro and small business legislation, and disarmament proposals.

In contrast, the measure of interest metric in the context of the Federal Government is based on the frequency of normative acts citations in the Official Gazette of the Federal Government (DOU).

This search was implemented via a data extractor of all contents of sections 2 and 3 from DOU since 2001, when its digital form became available. Acts located in section 2 deal with publications relating to public servants, such as appointments and designations of commissioned positions, while section 3 is meant to publish notices, contracts, amendments, cancellations, agreements, concessions, among others. Since section 1 is intended to publish normative acts itself, such as laws, decrees, resolutions, normative instructions, ordinances, and other normative acts of general interest, this session is not used to quantify the citations of normative acts in itself.

After extraction, the text content is structured, the citation frequency for each normative act is calculated and then normalized to obtain a metric value between 0 and 100.

As a result, Fig 10 shows the average interest for the 15 most popular normative acts on DOU. The most popular normative acts for the Federal Government are those related to public administration, such as rules that regulate bidding and contracts at the federal level, dispose of

| Normative act name and subject | Average interest |
| --- | --- |
| Law 13.982/2020 - COVID-19 emergency aid | 89.9 |
| Law 13.979/2020 - COVID-19 emergency measures | 49.8 |
| Law 13.467/2017 - Labor Reform | 36.9 |
| Law 13.146/2015 - Statute of People with Disabilities | 34.0 |
| S.L.[a]  123/2006 - Statute of Micro and Small Business | 31.8 |
| Law 11.343/2006 - Public Policies on Drugs | 27.8 |
| Law 10.826/2003 - Disarmament Statute | 25.7 |
| Law 8.112/1990 - Legal regime for federal civil servants | 23.0 |
| Law 8.666/1993 - Public bids and contracts | 22.8 |
| Law 11.101/2005 - Judicial recovery and bankruptcy | 22.1 |
| Law 12.799/2013 - Registration fee exemption for exams | 20.5 |
| Law 14.010/2020 - COVID-19 pandemic law | 20.4 |
| S.L. 101/2000 - Fiscal responsibility law | 19.1 |
| Law 12.016/2009 - Writ of Mandamus law | 18.0 |
| C.A.[b]  87/2015 - Interstate taxes and operations | 16.9 |

[a] Supplementary Law.
[b] Constitutional Amendment.

**Fig 9. Average interest of normative acts on Google Trends.**

| Normative act name and subject | Average interest | |
|---|---|---|
| Law 8.666/1993 - Public Adm bids and contracts | 100.0 | |
| Law 8.443/1992 - Federal Audit Court | 94.6 | |
| Law 8.112/1990 - Regime for servants of the Union | 80.5 | |
| Law 10.520/2002 - Law of the Auction | 49.5 | |
| Law 13.303/2016 - State Law | 46.1 | |
| Law 10.887/2004 - Retirement of the public servants | 41.0 | |
| C.A. 41/2003 - Social security of the public servants | 33.3 | |
| C.A. 47/2005 - Retirement of the public servants | 24.4 | |
| Law 11.416/2006 - Judicial servants careers | 19.2 | |
| Law 13.135/2015 - Pension in case of death | 17.5 | |
| S.L. 123/2006 - Statute of Micro and Small Business | 13.9 | |
| Law 12.772/2012 - Federal Magisterial careers plan | 10.2 | |
| Decree 7.892/2013 - Price Registration System | 8.1 | |
| Decree 5.450/2005 - Federal auction purchases | 7.7 | |
| Law 13.979/2020 - COVID-19 emergency measures | 7.2 | |

**Fig 10. Average interest of normative acts on DOU.**

internal rules for the Union's Court of Auditors, and establish the legal regime of the public civil servants of the Union.

It is important to mention that the values of the two measures of interest are not comparable since the data from Google Trends do not represent the frequency of citations of a normative act but rather the result of a Google calculation method that considers factors such as geography. In addition, both metrics are not calculated over the same time interval, making them incomparable among themselves.

**Linguistic complexity.** The last metric of interest is related to the complexity of each regulation. It is relevant for several reasons, as more complex regulations may force regulated entities to employ more personal and spend more time understanding them. Moreover, it can also make it less accessible to the general public as the language gets too specific.

Inspired by [17], we employ three different metrics to compare regulations' complexity. The first metric is the median sentence length. The median is used to avoid the effects of outliers that can appear when parsing sentences in law documents, such as tables or other bodies of text. The main assumption is that longer sentences tend to be more challenging to understand and, consequently, increase the document's complexity.

The second metric employed is Shannon's entropy, a measure of the average information of a single message from a given source [67]. It can be interpreted as measuring the frequency that new ideas (or words) are introduced in documents. Consequently, simpler and more focused documents have a lower entropy score than more complex documents. The entropy can be defined as

$$H(X_j) = -\sum_{i=1}^{N} p(x_{i,j}) log_2(p(x_{i,j})),$$ (3)

in which $X_j$ denotes the $j$-th document, $p(x_{i,j})$ indicates the probability/frequency of the word $x_{i,j}$ occurring in document $j$ and $N$ is the total number of words in document $j$.

The final metric is the frequency of conditional words in the text, which counts the number of branching words (in English, those are words such as "if", "but", and "provided") found in any given part in the text. Since we are working with Portuguese text, we adapted the

conditional terms to words that denote a similar branching idea, including the Portuguese words *"se"*, *"caso"*, *"quando"*, *"dado que"*, *"desde que"*, *"a menos que"*, *"a não ser que"*, *"embora"*, *"ainda que"* *"mesmo que"*, *"posto que"* and *"em que"*.

The goal of evaluating these three metrics is to understand if regulations are getting more complex by analyzing if the ideas are extended and wordy. Recall that the following analysis is based on the classification results obtained in Section Results and Discussion. Therefore, the accuracy of the results is directly related to the quality of the classification procedure.

The Complexity metrics considered in this work are not directly related. In other words, they express the complexity of a document by measuring different dimensions. As a consequence, one metric can increase over time while the other two may present different behaviors. Thus, to assess if the regulations from a sector became more or less complex, we have to analyze the three metrics together.

For example, if the number of conditional words for a given sector increases, the entropy can decrease if the total number of words is the same as the idea is being presented using a smaller word variation.

The main goal of the complexity analysis is to understand how it evolved over time. We can analyze it on a macro-level by grouping all sectors and calculating the median for each metric over time. Moreover, in order to have comparable results, we standardized each metric (zero mean and unit variance) and shifted them by subtracting the first value of the series, so each metric starts at zero. The results are shown in Fig 11.

It is evident that the overall complexity increased in the observed period. The conditional count metric was reduced starting in the middle of the 1990s, but its value is still higher than the beginning of the period. On the other hand, both entropy and median sentence length increased substantially starting in 2000.

However, it is essential to note that different sectors of the economy can present different regulatory dynamics. Thus, the overall complexity might not present the whole picture of a sector regulation over time. Moreover, since each sector would have three curves representing different metrics, the complexity trend could end up being challenging to present. In order to represent each sector as a single time-series, we employed kernel principal component analysis [68] (KPCA) to transform the three metrics into a one dimensional measure.

Since the metrics are not necessarily correlated, the same is true with respect to its one-dimensional projection. Therefore, we only present the sectors which have shown significant

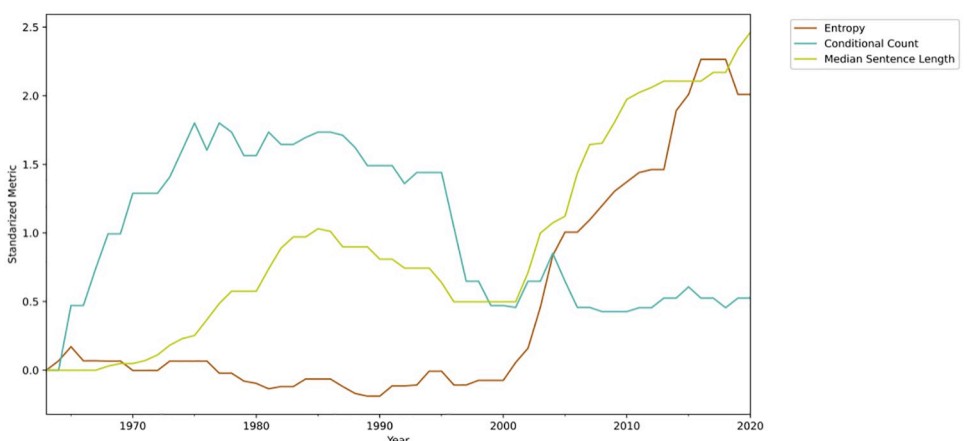

**Fig 11. Median metrics from all sectors grouped together, a moving average of 14 years was used to smooth out the curves.**

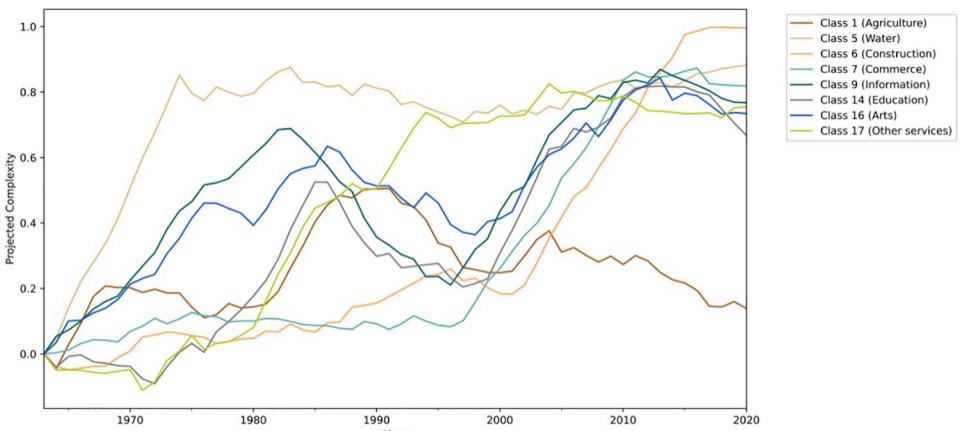

**Fig 12. Complexity projection, a moving average of 14 years was used to smooth out the curves.**

levels of correlation between the KPCA projection and their complexity metrics, which implies that an increase in the projected complexity is related to an increase in the complexity metrics. From all classes, there were eight that satisfied this requirement and their projections are presented in Fig 12.

Most of the observed sectors had an increase in regulation linguistic complexity throughout the period from 1964 to 2020. There are two distinct moments where regulations from most sectors become more complex. First around 1970 and then in the year 2000. Since the curves were smoothed using a moving average, the variations are not reflected instantly. So these two periods can be attributed to the new form of government established in 1964 and the re-democratization that started in 1985.

As stated before, the results presented in this section should not be considered as facts. They are based on classifications from a model that can (and most likely will) make mistakes, and on noisy metrics that try to represent the complexity of a text. Instead, they can be used as a decision support tool for policy-makers; in other words, they can help gain insights on how regulations evolved over time and how they can be improved.

In terms of tool maintenance, the processes of extracting and aggregating the regulations, classifying them into economic sectors and generating the metrics and visualizations is automated using cloud services and workflow management tools that are hosted and maintained by the Data Science Coordination team from the National School of Public Administration (ENAP). The RegBR's results, datasources and relevant analysis are presented on the Infogov's data portal https://infogov.enap.gov.br/regbr for public consultation and are also shared with the regulatory agencies that show interest.

## Conclusion

This paper presents RegBR, an active transparency framework that enables and facilitates regulatory analysis and monitoring of relevant legislative metrics over time. It also includes a novel benchmark for text classification of Brazilian federal normative legislation into economic sectors since 1964, using data gathered from decentralized sources and classified using state-of-the-art natural language processing models.

RegBR implemented different metrics to evaluate the Brazilian regulatory stock, such as text linguistic complexity, law measure of interest, law restrictiveness, and citation relevance of each industry, all of them tracked over time. Understanding those factors and how they evolve

are essential to better identify and prioritize regulations that may need reforms, besides being a useful tool for policymakers to measure their own work. Also, the metrics available can be used to subsidize new regulatory studies based on RegBR data and to serve as comparison by future studies that wishes to analyze government changes and their relationship with federal regulations produced by the legislative branch.

Another important contribution of the present work is the compilation of a centralized database with the aggregation of different regulatory acts from 1891 to the present day, which will be made available to researchers and professionals for further investigation, enabling transparency and reducing future costs for obtaining data and disseminating information.

The authors hope that from this rich corpus of information, new machine learning techniques can be used to train better models, which will improve the automatic classification of regulations in different economic sectors. Improving the classification step can boost the quality and accuracy of the subsequent metrics. In addition, we expect to keep increasing the number of advance metrics gathered by the project as well as increasing the types of legislation analyzed. Furthermore, it should be interesting consider tracking some economic effects in order to follow from regulatory changes when applied to different sectors. Such framework should be also replicated in other Portuguese-speaking countries, and our proposed metrics should be adapted to many different languages and legal corpora as well, possibly allowing for a world wide comparison of regulatory evolution.

## Supporting information

**S1 Table. Classification final table with its sections and divisions.**
(PDF)

## Acknowledgments

The authors wish to thank the National School of Public Administration (ENAP) for all the support and partnership in the development of the RegBR project. We also would like to thank the relevant feedback received by the regulatory agencies, which contributed with clear ideas so that the project could be used in practice by policy makers.

## Author Contributions

**Conceptualization:** Letícia Moreira Valle, Leonardo Monteiro Monasterio.

**Data curation:** Letícia Moreira Valle, Stefano Giacomazzi Dantas, Daniel Guerreiro e Silva.

**Formal analysis:** Letícia Moreira Valle, Stefano Giacomazzi Dantas.

**Methodology:** Letícia Moreira Valle, Stefano Giacomazzi Dantas, Daniel Guerreiro e Silva.

**Project administration:** Ugo Silva Dias.

**Resources:** Leonardo Monteiro Monasterio.

**Supervision:** Daniel Guerreiro e Silva, Ugo Silva Dias, Leonardo Monteiro Monasterio.

**Validation:** Letícia Moreira Valle, Stefano Giacomazzi Dantas, Daniel Guerreiro e Silva, Leonardo Monteiro Monasterio.

**Visualization:** Letícia Moreira Valle, Stefano Giacomazzi Dantas.

**Writing – original draft:** Letícia Moreira Valle, Stefano Giacomazzi Dantas, Daniel Guerreiro e Silva.

**Writing – review & editing:** Letícia Moreira Valle, Stefano Giacomazzi Dantas, Daniel Guerreiro e Silva, Ugo Silva Dias, Leonardo Monteiro Monasterio.

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
