## [Decision Letter · Decision Letter 0]

10 Aug 2022

PONE-D-22-04202RegBR: A novel Brazilian government framework to classify and analyze industry-specific regulationsPLOS ONE

Dear Dr. Valle,

Thank you for submitting your manuscript to PLOS ONE. After careful consideration, we feel that it has merit but does not fully meet PLOS ONE’s publication criteria as it currently stands. Therefore, we invite you to submit a revised version of the manuscript that addresses the points raised during the review process.

We look forward to receiving your revised manuscript.

Kind regards,

Sathishkumar V E

Academic Editor

PLOS ONE

Journal Requirements:

2 Thank you for stating in your Funding Statement:

“This research was partially supported by the National School of Public Administration – ENAP, Government of Brazil (Grant TED UnB-ENAP 83/2018).”

“The authors wish to thank the National School of Public Administration (ENAP) for all the support and partnership within the development of the RegBR project, providing funding, infrastructure and establishing contact with regulatory agencies. We also would like to thank the relevant feedback received by the Brazilian regulatory agencies, who contributed with clear ideas so that the project could be used in practice by police makers”

“This research was partially supported by the National School of Public Administration – ENAP, Government of Brazil (Grant TED UnB-ENAP 83/2018).”

Reviewers' comments:

Reviewer's Responses to Questions

**Comments to the Author**

1. Is the manuscript technically sound, and do the data support the conclusions?

Reviewer #3: No

Reviewer #4: Yes

2. Has the statistical analysis been performed appropriately and rigorously? 

Reviewer #3: Yes

Reviewer #4: Yes

3. Have the authors made all data underlying the findings in their manuscript fully available?

Reviewer #3: Yes

Reviewer #4: Yes

4. Is the manuscript presented in an intelligible fashion and written in standard English?

Reviewer #3: Yes

Reviewer #4: Yes

5. Review Comments to the Author

Reviewer #3: RegBR: A novel Brazilian government framework to classify and analyze industry specific Regulations

Reviewer Comments:

In this paper, the authors apply machine learning, deep learning and transfer learning models along with word weighting and embedding schemes to generate framework to classify industry specific regulations. The various combinations of techniques used by the authors are appreciable. The paper is well organized and written. The interpretation and description of the experimental results are also explained clearly. However, this manuscript have some weak points, it should be further improved before consider for publication. Some of my observations are

1. Figures 1,2,3,7 need to be drawn clearly and explanation about each figure to be incorporated in corresponding section.

2. In figure 5, x-axis legend is not mentioned and proper explanation is required.

3. In figure 9, only 8 classes are displayed? What about other classes. Justification is required.

4. Why validation is not performed? Justify

5. Why the authors specifically choose CNN and LSTM classifiers?

6. According to the results obtained, if statistical models perform better than machine, deep and transfer learning models, then what is the use of these latest techniques for analysis? Is hypertuning of these models not done properly? Justification is required.

Reviewer #4: Manuscript needs English language check as the sentence formation errors occur throughout the manuscript

What strategy is used to select hyperparameters?

Explain the dataset used in detail. Make a table to summarize the dataset features.

Explain the proposed work in detail

Elaborate Normative acts.

Application of the proposed system should be explained in detail.

What is the need of using ensemble classifier?

Whole manuscript should be reorganised following standard scientific article requirements

6. PLOS authors have the option to publish the peer review history of their article (what does this mean?). If published, this will include your full peer review and any attached files.

Reviewer #3: No

Reviewer #4: **Yes: **Usha Moorthy

---

## [Author Response · Author response to Decision Letter 0]

2 Sep 2022

Dear Prof. Sathishkumar V E,

On behalf of all the authors of the manuscript "RegBR: A novel Brazilian government framework to classify and analyze industry-specific regulations," we would like to thank you very much for revising our manuscript and for the relevant comments provided by you and the two reviewers, which helped us to improve our work. We submit our revised manuscript for appreciation and also a point-by-point response to the reviewer's comments.

As suggested in the decision e-mail, we also decided to modify the funding statement in order to suit the journal's requirements. The new funding statement is as follows:

This research was supported by the National School of Public Administration – ENAP, Government of Brazil (Grant TED UnB-ENAP 83/2018). There was no additional external funding received for this study.

For the acknowledgements section of our manuscript, we also like to slightly modify the statement to the following one:

The authors wish to thank the National School of Public Administration (ENAP) for all the support and partnership in the development of the RegBR project. We also would like to thank the relevant feedback received by the regulatory agencies, which contributed with clear ideas so that the project could be used in practice by policy makers.

We hope that the improvements we have made to this paper will be enough to enable it to be considered for publication by this renowned journal.

Thank you for considering our manuscript.

Best regards,

The authors

 Academic editor:

https://journals.plos.org/plosone/s/file?id=wjVg/PLOSOne_formatting_sample_main_body.pdf and https://journals.plos.org/plosone/s/file?id=ba62/PLOSOne_formatting_sample_title_authors_affiliation s.pdf

R. We have made changes to figures, tables, and author affiliations formatting and we now confirm that our manuscript meets PLOS ONE’s style requirements. We used PACE

digital diagnostic tool to ensure that our Figures meet PLOS requirements.

2 Thank you for stating in your Funding Statement:

“This research was partially supported by the National School of Public Administration – ENAP, Government of Brazil (Grant TED UnB-ENAP 83/2018).”

R. We would like to change our funding statement to the following:

“This research was supported by the National School of Public Administration – ENAP, Government of Brazil (Grant TED UnB-ENAP 83/2018). There was no additional external funding received for this study.”

“The authors wish to thank the National School of Public Administration (ENAP) for all the support and partnership within the development of the RegBR project, providing funding, infrastructure and establishing contact with regulatory agencies. We also would like to thank the relevant feedback received by the Brazilian regulatory agencies, who contributed with clear ideas so that the project could be used in practice by police makers”

“This research was partially supported by the National School of Public Administration – ENAP, Government of Brazil (Grant TED UnB-ENAP 83/2018).”

R. We also would like to change our Acknowledgments section to the following:

The authors wish to thank the National School of Public Administration (ENAP) for all the support and partnership in the development of the RegBR project. We also would like to thank the relevant feedback received by the regulatory agencies, which contributed with clear ideas so that the project could be used in practice by policy makers.

R. We have confirmed that our reference list is complete and correct.

Reviewers' comments:

● Reviewer #3

In this paper, the authors apply machine learning, deep learning and transfer learning models along with word weighting and embedding schemes to generate framework to classify industry specific regulations. The various combinations of techniques used by the authors are appreciable. The paper is well organized and written. The interpretation and description of the experimental results are also explained clearly. However, this manuscript have some weak points, it should be further improved before consider for publication. Some of my observations are

R. The modifications associated with reviewers' #3 comments are highlighted in green in the Revised Manuscript with Track Changes document. Next, we provide a point-by-point response to the reviewers' #3 comments.

1. Figures 1,2,3,7 need to be drawn clearly and explanation about each figure to be incorporated in corresponding section.

R. Thank you for this comment. We have redrawn the indicated Figures and added a clearer explanation in the corresponding sections. In order to adapt our work to the journal's style format, some tables have become figures and that's why the new figures in this comment are now the Figures 2, 3, 4 and 8.

 2. In figure 5, x-axis legend is not mentioned and proper explanation is required.

R. Thank you for noticing that. We adjusted the image adding the x-axis legend and its scale. Also, we added a paragraph explaining the image and an example of its interpretation. Fig 5 is now named as Fig 6.

3. In figure 9, only 8 classes are displayed? What about other classes. Justification is required.

R. We appreciate the question. In response, we only displayed the classes that presented a significant correlation level between their KPCA projection and their complexity metrics. A total of 8 classes satisfied these conditions. We made it clearer in the manuscript, at the subsection Linguistic complexity right before Fig 9.

4. Why validation is not performed? Justify

R. Thank you for bringing up this point. In fact, RegBR is the first Brazilian regulatory framework that creates centralized regulatory metrics, and for this reason, there is no similar work to be compared with. On the other hand, we can correlate the observed results with known episodes of regulatory change in Brazilian history, such as the time of the creation of the regulatory agencies, the implementation of new economic plans or the energy crisis that happened in the 2000s, as related in the Regulatory metrics conception subsection. Also, after presenting the results to the national regulatory agencies, we received feedback in the sense that our metrics results corroborated with internal metrics created by the agencies. An example of this fact is that the national water agency (ANA), which uses a plain language approach to monitor the linguistic complexity of their texts, informs us that they were expecting ANA’s agency to have one of the biggest linguistic complexity within the sectors for the types of normative acts analyzed in our work, which was confirmed by our linguistic complexity metric.

5. Why the authors specifically choose CNN and LSTM classifiers?

R. Thank you for the question. The reasons for choosing CNN and LSTM are based on two main factors. First, they are often used in many NLP applications. Secondly, before the creation of the Transformer architecture, they were used as building blocks for some state-of-the-art NLP models. These two points led us to include models based on these two neural network designs to our benchmark. We added a paragraph in the manuscript to make that choice clearer.

6. According to the results obtained, if statistical models perform better than machine, deep and transfer learning models, then what is the use of these latest techniques for analysis? Is hypertuning of these models not done properly? Justification is required.

R. Thank you for raising this point. We added more information in the text regarding the importance of benchmarking different models. For some NLP tasks, such as translation, there are standard datasets that are used to evaluate models. In that case, comparing the proposed model with the current state-of-the-art is enough most of the time. However, in our application, there were no past performances to serve as a baseline to indicate what would be considered a "good" model. Thus, we performed an evaluation of different models and approaches. With regard to the hypertuning, we used grid-search with the values for the parameters that are described in the literature and we have updated this information in the manuscript. Moreover, in the text, we raised some possible reasons why the transfer-learning methods presented underwhelming results.

Once again, we thank the reviewer for devoting his/her time to assess our manuscript and for the helpful comments which have guided us to enrich the original manuscript. We hope that our manuscript and revision of the manuscript are satisfactory.

● Reviewer #4

R. The modifications associated with reviewers' 4 comments are highlighted in orange in the Revised Manuscript with Track Changes document. Next, we provide a point-by-point response to the reviewers' 4 comments.

1. Manuscript needs English language check as the sentence formation errors occur throughout the manuscript

R. We performed a manuscript language review and corrected grammar and sentence formation errors. We hope that this revised version of the manuscript will meet the language requirements.

2. What strategy is used to select hyperparameters?

R. Thank you for your question. We made it clearer in the paper how we select the hyperparameters. We performed a grid-search using the hyperparameter values often used in the literature for this kind of task.

3. Explain the dataset used in detail. Make a table to summarize the dataset features.

R. Thank you for the comments. In the text classification task, all the features were derived from the normative act texts themselves. To summarize the data workflow, first we build an ETL (Extract, transform, load) system to gather normative act texts from various sources and centralize them in a database. Then, we prepared the data by removing stop words, malformed characters and numbers, and standard sentences and headers that appear in all normative texts. Lastly, we transform text to numeral representation in different ways depending on the model used, like using tf-idf, word embeddings, or in the case of transformer-based models, the specific encoder proposed by the model architecture.

More details on each model's feature extraction approach are available in the subsection Details on Text Classification.

4. Explain the proposed work in detail

R. Our paper proposes a novel regulatory framework applicable to Brazil with the goal of producing relevant information on the national regulatory situation in order to: promote active transparency of public information; subsidize new regulatory studies based on RegBR data; and assist decision makers to measure what their organization produces in terms of volume and characteristics of regulations. For the first time, we classify normative acts in industry-specific classes and create a set of new regulatory metrics that concern linguistic complexity, restrictiveness, legal interest, and industry-specific citation relevance. We made some adjustments in the manuscript such that the goals and applications of the proposed work are better explained.

5. Elaborate Normative acts.

R. By a dictionary definition, a normative act means any law, decree, resolution, regulation, administrative direction, instruction, rule, ordinance, or other decision that creates legal consequences for more than one individual. We also added this definition in the manuscript, in the Normative Acts data structure subsection.

6. Application of the proposed system should be explained in detail.

R. Thank you for raising this point. We are producing a second paper focused exclusively on RegBR applications, but in order to make the applications clear in this article, we added a new paragraph in the Introduction section explaining in more detail the possible applications of the system.

7. What is the need of using ensemble classifier?

R. Ensemble models tend to yield better results when there is a significant diversity among the models. In our case, we combine a ridge classifier, which is a classifier that converts the task into a regression and generates labels accordingly, and an SVM classifier. These two models have a few complementary strengths and weaknesses, and when combined, generate a more precise and robust prediction.

8. Whole manuscript should be reorganised following standard scientific article requirements

R. Parts of the manuscript text, figures, and tables have been rewritten so we could meet all PLOS ONE organization and style requirements available in https://journals.plos.org/plosone/s/file?id=wjVg/PLOSOne_formatting_sample_main_b ody.pdf. We hope that the improvements we have made to this paper should be enough to enable it to be published by this renowned journal. We remain open to modifying specific parts of the manuscript that you feel don't meet the journal requirements.

We would like to thank the reviewer for providing some helpful comments and suggestions that have improved our manuscript. We hope that our responses and revisions of the manuscript are satisfactory.

---

## [Decision Letter · Decision Letter 1]

13 Sep 2022

RegBR: A novel Brazilian government framework to classify and analyze industry specific regulations.

PONE-D-22-04202R1

Dear Dr. Valle,

We’re pleased to inform you that your manuscript has been judged scientifically suitable for publication and will be formally accepted for publication once it meets all outstanding technical requirements.

Kind regards,

Sathishkumar V E

Academic Editor

PLOS ONE

Additional Editor Comments (optional):

Reviewers' comments:

Reviewer's Responses to Questions

**Comments to the Author**

1. If the authors have adequately addressed your comments raised in a previous round of review and you feel that this manuscript is now acceptable for publication, you may indicate that here to bypass the “Comments to the Author” section, enter your conflict of interest statement in the “Confidential to Editor” section, and submit your "Accept" recommendation.

Reviewer #2: All comments have been addressed

Reviewer #4: (No Response)

2. Is the manuscript technically sound, and do the data support the conclusions?

Reviewer #2: Yes

Reviewer #4: (No Response)

3. Has the statistical analysis been performed appropriately and rigorously? 

Reviewer #2: Yes

Reviewer #4: (No Response)

4. Have the authors made all data underlying the findings in their manuscript fully available?

Reviewer #2: Yes

Reviewer #4: (No Response)

5. Is the manuscript presented in an intelligible fashion and written in standard English?

Reviewer #2: Yes

Reviewer #4: (No Response)

6. Review Comments to the Author

Reviewer #2: Accepted, the author has incorporated all the comments. In its current form the paper ua ready for publication

Reviewer #4: (No Response)

7. PLOS authors have the option to publish the peer review history of their article (what does this mean?). If published, this will include your full peer review and any attached files.

Reviewer #2: No

Reviewer #4: **Yes: **Usha Moorthy

---

## [Editor Report · Acceptance letter]

19 Sep 2022

PONE-D-22-04202R1 

RegBR: A novel Brazilian government framework to classify and analyze industry-specific regulations 

Dear Dr. Moreira Valle:

I'm pleased to inform you that your manuscript has been deemed suitable for publication in PLOS ONE. Congratulations! Your manuscript is now with our production department. 

Kind regards, 

on behalf of

Dr. Sathishkumar V E 

Academic Editor

PLOS ONE